# Congenital Pseudarthrosis of the Clavicle in Children: A Systematic Review

**DOI:** 10.3390/children9020147

**Published:** 2022-01-24

**Authors:** Alessandro Depaoli, Paola Zarantonello, Giovanni Gallone, Giovanni Luigi Di Gennaro, Daniele Ferrari, Leonardo Marchesini Reggiani, Aniello Manca, Giovanni Trisolino

**Affiliations:** Department of Pediatrics Orthopedics and Traumatology, IRCCS Istituto Ortopedico Rizzoli, 40136 Bologna, Italy; alessandro.depaoli@ior.it (A.D.); giovanni.gallone@ior.it (G.G.); giovanniluigi.digennaro@ior.it (G.L.D.G.); daniele.ferrari@ior.it (D.F.); leonardo.marchesinireggiani@ior.it (L.M.R.); aniello.manca@ior.it (A.M.); giovanni.trisolino@ior.it (G.T.)

**Keywords:** congenital, pseudarthrosis, nonunion, clavicle, cleidocranial dysostosis, child, rare disease, bone graft

## Abstract

(1) Background: Congenital pseudoarthrosis of the clavicle is a rare condition due to the failure of the union process of the ossification nuclei of the clavicle. The aim of this study was to conduct a systematic review of relevant case series about the argument to find an up-to-date base of evidence for treatment choice. (2) Methods: an electronic literature research of Ovid, MEDLINE and the Cochrane Library databases was conducted, and articles were selected based on inclusion criteria. Demographic data, clinical features, treatment options, outcomes and complications were analyzed. (3) Results: 21 articles met the inclusion criteria, showing a poor overall study quality; 231 pediatric patients (240 clavicles) were analyzed. The condition was typically right sided, showed no sex predominance and no clear predisposing factors. 156 patients underwent surgical treatment, mainly open debridement and refresh of bony ends, fixation with pin or plate and bone graft, with a successful union rate of 87.4%. The nonunion rate was significantly higher in the allograft group (44.4%, *p* = 0.019). (4) Conclusions: this paper presents an updated systematic review about treatment of congenital pseudoarthrosis of the clavicle. We confirm the generally satisfactory results of surgery, demonstrating that successful union is achievable in 87.4% of cases with a prevalence of 15.7% of major complications. Nonetheless our results should be interpreted with caution due to several limitations.

## 1. Introduction

Congenital pseudarthrosis of the clavicle (CPC) is a rare condition typically diagnosed during the first few years of life, due to the failure of the union process of the ossification nuclei of the clavicle [1,2]. The first documented case of CPC, accurately described by Fitzwilliams in 1910, was included in a case series of cleido-cranial dysostosis as a monostotic variant [3]. This condition seems more common in females and typically unilateral [4,5]. Right clavicle is much more affected than the contralateral; when found on the left side, some authors hypothesized a potential association with dextrocardia [2]. Bilateral involvement occurs in 10% of cases. Although the etiology remains unknown, the pathogenesis is probably related to the embryology of the clavicle [6,7,8]. There was at least one well documented family with several members affected by CPC and a case of CPC in two twins [9], but a clear genetic pattern has not yet been associated with this condition [1,2,9,10,11]. The CPC is present at birth, but it is most often identified between the first months and the 5 years of life [5]. Patients present a painless swelling over the middle third of the clavicle, which tends to increase with growth. Differential diagnoses of CPC include: obstetric fracture (which tends to rapidly heal with an exuberant callus), cleido-cranial dysostosis, neurofibromatosis and post-traumatic nonunion [12]. Patients may remain asymptomatic during the entire life and the shoulder’s range of motion is usually normal and painless [7,13,14,15,16,17]. For these reasons, indication for surgery, timing to surgery and surgical technique remain controversial [13,14]. The aim of this study was to conduct a systematic review of all relevant studies regarding CPC, to assess which treatment is offered for this condition in the most relevant case series published in literature, to retrieve demographic and clinical data and to derive from them any correlation between various influencing factors, outcomes and complications to try to find an up-to-date base of evidence for treatment choice.

## 2. Materials and Methods

### 2.1. Study Design and Search Strategy

This review was performed in accordance with the format of the Preferred Reporting Items for Systematic Reviews and Meta-Analyses (PRISMA) [18]. An electronic research in Ovid, MEDLINE and the Cochrane Library databases was conducted in December 2020 by two observers (P.Z., G.G) using the terms “pseudarthrosis”, “pseudoarthrosis”, “nonunion”, “non-union”, “clavicle”, “congenital”. The research was then replicated using the appropriate MeSH terms. We aimed to keep our search as comprehensive as possible to increase our search results. The searching process was not limited by year of publication, type of journal or level of evidence (LOE). Bibliographies were evaluated in order to find other relevant studies.

### 2.2. Eligibility Criteria

The final selection included original articles that addressed (1) original articles about children (<18 years) with CPC; (2) in English; (3) reporting conservative and/or surgical management; (4) involving three or more cases; (5) peer reviewed. The exclusion criteria were: (1) studies not reporting original research, including review articles, expert opinion, current concepts articles, posters or abstracts at annual meetings or masters’ theses without subsequent peer-reviewed publication of an article; (2) studies about animals; (3) articles written in languages different from English; (4) studies not reporting CPC management; (5) studies reporting only adult cases (>18 years); (6) case reports with less than 3 cases. Furthermore, we screened case reports with less than 3 cases to report rare events and major complications, in order to avoid the overestimation of positive results in this review.

### 2.3. Article Selection

Initially, two non-blinded authors (P.Z. and G.G.) reviewed all titles and abstracts to discard studies that undoubtedly did not meet one or more of the above criteria. If a study met all criteria or the abstract did not provide enough information to include or exclude the report, full texts were obtained and further evaluated for data extraction. No disagreement about study inclusion was observed between the two reviewers.

### 2.4. Data Extraction and Analysis

Two authors (P.Z. and G.G.) independently analyzed relevant articles retrieved with the systematic search. They manually extracted data from full texts compiling an Excel spreadsheet (Microsoft, Redmond, Washington DC, USA), while a third author (A.D.) checked agreement and accuracy of databases to minimize subjective evaluation. The following items were collected from each study and inserted into a predefined table: first author, publication year, demographics and clinical features (patient’s sex, affected side, age at presentation, family history, associated medical conditions, presence of anomalies of the cervical spine such as cervical ribs, prominent transverse processes, abnormality of the cervico-thoracic junction), presence of symptoms, aspects related to the treatment (surgical or conservative option); in case of surgical treatment: patient’s age at surgery, characteristic of the surgical procedure, kind and length of postoperative immobilization, type of rehabilitation, length of the follow-up period, complications and outcomes. Complications were rated according to the Clavien-Dindo-Sink (CDS) Classification [19,20]; in particular, we diversified each complication based on the necessity of additional medical and/or surgical treatment. Minor complications included conditions that have no or minimal clinical relevance, requiring no other medical treatments or causing little deviation from the normal postoperative course (grade 1 or 2), i.e., superficial infection and unslightly scarring. Major complications included those conditions requiring surgery, such as deep infection, permanent brachial plexus lesion, hardware breakage or failure, recurrent nonunion and/or surgical demanding fracture (grade 3) or conditions that were limb-threatening, life-threatening (grade 4) or causing the death of the patient (grade 5). Hardware removal after consolidation and multistage procedures were not considered as complications. When applied, we reported the clinical outcomes scores. Post-operative scores, assessing the clinical and radiographic union of the pseudarthrosis, were considered clinician-derived outcomes, whereas patient-reported scores, such as the Disabilities of the Arm, Shoulder and Hand (DASH) outcome questionnaire [21], the Patient-Reported Outcomes Measurement Information System (PROMIS^®^) [22], the Constant-Murley Score (CMS) [23,24,25]. The visual analogue scale (VAS) was considered patient-derived outcome [26].

### 2.5. Quality Assessment and Statistical Analysis

The quality of the studies retrieved in the systematic review was assessed according to the Oxford Center for Evidence-Based Medicine (CEBM) [27] and the Modified Coleman Methodology Score (mCMS) [28]. Descriptive analysis using means and ranges (minimum, maximum, 95% confidence interval) of the pooled data across the included studies was performed. Normality was tested using Pearson’s Chi-squared test or Fisher’s exact test were used respectively in big and small groups to compare categorical variables (e.g., union and nonunion) between different surgical groups, Kolmorogov-Smirnov test was applied for continuos variables. Spearman’s and logistic correlations were used to correlate demographic details (e.g., age) with outcomes of interest (e.g., complications). A *p*-value of less than 0.05 was considered to be significant. All calculations were performed using Microsoft Excel 16.46 and IBM SPSS Statistics 27.0.1.0.

## 3. Results

From an initial pool of 2307 potentially eligible studies, a total of 21 studies (240 CPC in 231 children) were retrieved for the final analysis. Additionally, 53 cases of CPC were found by screening of case reports in English Literature; of them 46 were pediatric cases and 19 of them were treated surgically (details in Appendix A). The inclusion process is illustrated in Figure 1.

In Table 1 we reported the characteristic of the included studies. All studies, except three [7,13,29], were case-series, classified as CEBM level 4; the others were retrospective comparative studies, graded as level 3b. All studies were published between 1963 and 2020. The mean mCMS was 23 points (range 0–50), highlighting poor methodological quality across all studies.

### 3.1. Demographics and Clinical Characteristics

The side of the lesion was not specified in 11 of these patients (4.8%) [29]. In 92.7% of the remaining cases, CPC lied on the right side (204/220), in 3.2% on the left (7/220) and in 4.1% it was bilateral (9/220). Patient’s gender was specified in 197 patients (85.3%). The male/female overall ratio was apparently unbalanced (89/108). However, by distributing the M:F ratio by decades of publication of the studies (Figure 2), we observed a clear trend to 1:1 ratio. For five patients, a family history of CPC was reported [1,7,13]. 

The mean age at which the caregiver (e.g., parents, obstetrician, family doctor) noted the deformity for the first time was reported in 89 patients and averaged 2.1 years, while the mean age at which the case was referred to an orthopaedic surgeon was reported for 133 patients and averaged 4.6 years (range 0–16). Associated medical conditions were reported in 26 cases (26/231, 11.3%) [1,13,14,15,16,17,34,38]; of these only ten cases were better specified: one cervical spina bifida occulta [15], two cases with prominent forehead [15], one Smith-Lemli-Opitz syndrome [34], one deletion of the chromosomes 3 and 7 [38], one multiple tetramelic deformity [17], one funnel chest [15], one case of anoxia at birth with no other specified consequences[1] and one patient with 3 café-au-lait spots (the diagnosis for neurofibromatosis was excluded) [16]. Overall, no cases of dextrocardia were found. Radiographic abnormalities of the cervico-thoracic junction were reported in 9 patients (3.9%): six cases with cervical ribs [15,16,31], three patients with a C7 hypertrophic transverse process [38]. Clinical evaluation was reported in 181 cases. All patients referred aesthetic issues (i.e., swelling, prominence, deformity), while 40 patients (22.1%) suffered other symptoms, such as pain and/or impaired function. Pain and other symptoms showed a moderate but not statistically significant increase with age (β-coefficient 0.116, *p* = 0.093, graph in Appendix A). Most patients with symptoms were eventually treated with a surgical procedure (37/40 patients, 92.5%). Symptomatic presentation was significantly more frequent (*p* = 0.0003) among children undergoing surgery (29.1%) compared to patients treated conservatively (5.6%).

### 3.2. Treatment

75 patients (32.5%) underwent conservative treatment. None of the authors reported any detail concerning non-operative management and follow-up. 156 patients (67.5%) sustained surgical treatment (164 procedures). The most used surgical procedure consisted of open debridement, refresh of bony ends, fixation with pin or plate and bone graft (Table 2).

The collected data, grouped according to the surgical procedures, are shown in Table 2. The mean age at treatment was 5.9 years (range 0.5–16.6). The period of immobilization was reported in 99/164 procedures with an overall mean time of 7.1 weeks (range 2.0–18.6 weeks). Details are reported in Table 2.

### 3.3. Surgical Outcomes

The mean follow-up time was 46.1 months (range 0–214). Time to union was reported in only 33 cases and averaged 21.2 weeks (range 4–100). After excluding the 5 patients treated with pure excision, since this treatment did not pursue healing of the pseudarthrosis, successful union was finally achieved in 139/159 procedures (87.4%; 95% confidence interval: 82.3–92.6%), without any significant association with age at treatment (β-coefficient = −0.124, *p* = 0.156 graph in Appendix A) and surgical technique (*p*-values 0.278–1.0, details in Appendix A). Nonetheless, the nonunion rate was significantly higher in the allograft group in comparison with patients treated with autograft from the donor site (*p* = 0.019) (see details in Table 3). In addition to the 20 patients suffering nonunion, major complications were reported in 5 more cases [29,30,38,40]. Even by weighting each complication according to CDS classification, they showed no correlation with age at treatment (β-coefficient = 0.023, *p* = 0.830, see graphs in Appendix A) and no significant association with the surgical technique (*p*-values 0.093–1.0, details in Appendix A). Major complications other than nonunion were: two debridement-requiring infections [29,38], one hardware breakage [38], one skin irritation [40], one acute massive neuropraxia of brachial plexus 6 h postoperatively [30]. All these major complications were graded as 3 according to the CDS classification. No cases of refracture were reported in the pool of studies which met the inclusion criteria. Neither grade 4 nor grade 5 were found in the reviewed articles. By screening the case reports, we found 6 major complications and 1 minor complication in 19 surgical procedures. This major complication rate (6/19, 31.6%) did not differ significantly from the 15.7% found by this review (*p*-value = 0.085). One patient suffered refracture [42] and five suffered nonunion [42,43,44,45]; two of these nonunions were treated respectively with a vascularized fibular autograft [43] and a revision fixation covered with a flap of vascularized periosteum graft from medial femoral condyle [44]. The minor complication was a fracture on a screw hole 10 days after plate removal, which healed conservatively [46].

Post-operative symptoms were assessed in 126 patients [1,7,14,15,30,31,32,33,34,35,36,37,38,39,40,41], of them 10 patients referred pain and/or functional impairment. In particular, symptoms deterioration was observed in all five patients treated with simple pseudarthrosis excision (*p* = 0.018). Only three studies used validated clinical outcomes scores. Kim et al. compared the clinical outcomes of children’s cohort undergoing surgical treatment against children receiving non-operative treatment (respectively 11/24 patients and 5/23 patients) using the Quick-DASH Score and the PROMIS; they found difference in Quick-DASH but comparable achievements with PROMIS [13]. Studer et al. evaluated their 7 patients that underwent autograft and fixation with plate during follow-up with Constant-Murley Score: they found that all children had good functional outcome (score between 88 and 98) during postoperative period [39]. Haddad et al. measured post-operative pain in their three Masquelet procedures with the VAS; clinical outcomes were reported as good with no other details about score results [41]. 

## 4. Discussion

### 4.1. Epidemiology, Etiology, Clinical Presentation

CPC is a very rare condition and Literature keeps echoing the first conjectures presented almost 50 years ago, despite some new elements have slowly settled in the last decades. To date, CPC is a very rare congenital deformity of the shoulder girdle, with no gender predilection, that is generally noticed during the first two years of life. It is typically right-sided (>90%), while this research did not confirm the association between left side presentation and dextrocardia. Furthermore, no association was found with other congenital abnormalities of shoulder, neck and thorax. In fact, the overall prevalence of any abnormality of cervico-thoracic junction in the pool of patients of this study was less than 5%, in line with the estimated prevalence of anomalies of the thoracic outlet in general population [47]. Similarly, the familial inheritance or the association with genetic or chromosomic disorders is sporadic (less than 3% of cases). Therefore, based on the current evidence from literature, the etiology of CPC, along with its prevalence among population, remains unknown. From a clinical point of view, patients may present with painless swelling in the region of the clavicle with mobile ends of the clavicular segments at the site of pseudarthrosis. The deformity can become more obvious with development increasing the cosmetic concerns, while other symptoms such as pain, impaired function and limited range of motion are present in about one fourth of patients. This analysis showed that symptoms have an apparent slight increase with age, although we lack sufficient information regarding the natural history of the pathology during the adulthood, especially in untreated patients. While screening smaller case series of CPC, 5 cases of thoracic outlet syndrome (TOS) caused by CPC were found [42,48,49,50,51]: one of them was reported in a 15 year old adolescent[42], the other 4 patients were adults.

### 4.2. Surgical Treatment

Indications, timing for surgery and surgical technique are still debated between authors. Huntley proposed an evidence-based treatment reviewing most of the studies analyzed in our systematic review. He concluded that ‘in asymptomatic patients it is unclear if surgery confers benefits (justifying the risk) in the long term’; ‘if symptoms or deformity are progressive, surgery can be justified’; ‘bone graft is a useful component to the procedure over 4 years’ and ‘plate fixation rather than pinning in children older than 4 years’[52]. This study confirms the generally satisfactory results of surgery, demonstrating that successful union is achievable in 87.4% of cases with a prevalence of 15.7% of major complications (12.6% nonunions and 3.1% other major complications). Stable fixation with pins or plate and autograft is the most widely used technique, while we do not recommend simple excision or use of allograft alone, on the basis of our results. Age at surgery seems to be not relevant in terms of type of surgery, union rate and complication rate, although some authors recommended early surgery in order to perform less aggressive procedures [7,14,33,34]. Although no definitive evidence exists concerning the surgical outcomes in terms of pain resolution and function improvement, it has been reported that surgery can improve shoulder function and pain [13,39,41]. No postoperative fractures were described in the main pool of studies. Two cases of fractures were found during the screening of case report, but one of them involved a screw hole away from the pseudarthrosis site. Considering a mean follow-up of 46.1 months (range 0–214 months), refracture after healing of the pseudarthrosis site may be considered a very rare occurrence for at least the first 4 years after CPC treatment.

### 4.3. Limitations

Although this systematic review collected 21 case series about 231 patients with CPC, the rarity of this condition and the heterogeneity of treatments reported by different authors must be considered limitating factors. Most studies had a small number of cases: 7 studies presented 10 or more patients [1,7,13,15,29,30,38] and only 2 studies 30 or more patients [13,15]. Exclusion of articles not written in English might have concealed data from this analysis, perhaps important ones, but certainly not the most recent ones. All the studies included were retrospective case series spanning across 6 decades with mean mCMS of 23 (range 0–50) that indicated a poor overall methodological quality. All authors clearly stated prevalence of nonunions and complications, but the lack of raw data in some cases did not allow statistical inference of the entire pool of patients. The screening of case reports allowed including all available evidence on CPC and it could increase the power of analyses performed. These issues severely limit the strength of resultant evidence. However, as CPC is a very rare condition, high level studies could be unfeasible. Although the true effect is likely to diverge from effect estimates due to a very low level of evidence, we are confident that surgical treatment can be a safe and effective option, especially in symptomatic cases.

## 5. Conclusions

CPC is a rare, isolated condition, typically right-sided, with no sex predominance and no clear predisposing factors. In newborns, clavicle obstetric fracture (which rapidly heals), NF and cleido-cranial-dysostosis should be excluded. Parents ought to be reassured that CPC is a benign condition, but in children and adolescent it tends to slowly become more evident and, sometimes, symptomatic.

Surgical treatment is recommended in children with symptoms and/or very prominent deformity. In patients younger than 4 years, graft from local site and end-to-end bone suture should be the first choice of treatment. Over 4 years of age, surgical repair with pin or plate fixation and bone autograft is recommended. The surgical treatment achieved union in 87.4% of cases. However, due to the rarity of the condition this systematic review did not clarify what is the best age at treatment, indication for surgery, and what are the expected clinical, functional and cosmetic outcomes from a patient perspective. Therefore, the risks and benefits of surgery must be exhaustively discussed with children and parents. Further multicentric prospective studies and data from international registries are needed to fully understand natural history of CPC and to strengthen evidence that guide the management of this condition.

## Figures and Tables

**Figure 1 children-09-00147-f001:**
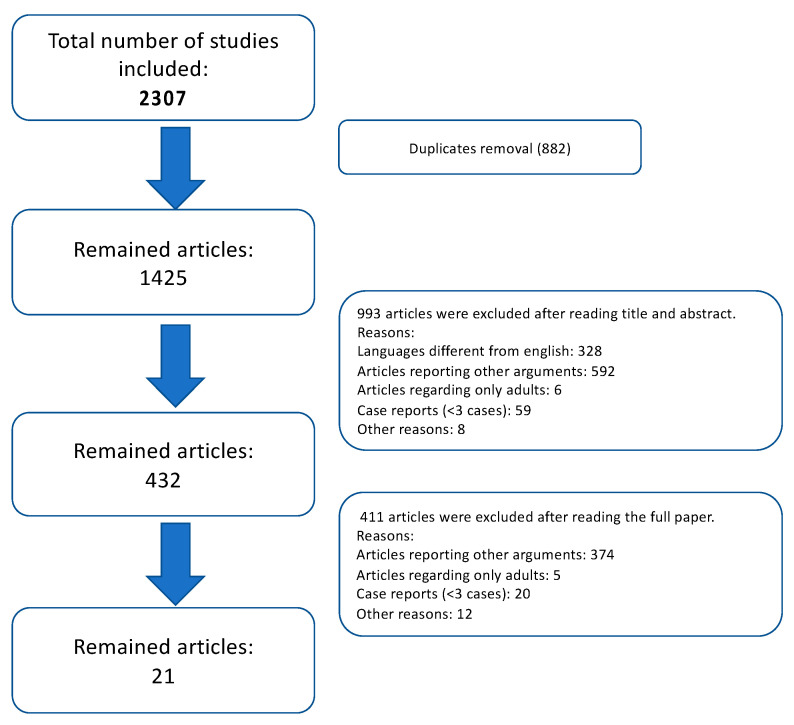
Flowchart highlighting the study acquisition details from all the articles found in Literature search to the pool of 21 articles which met all the inclusion criteria.

**Figure 2 children-09-00147-f002:**
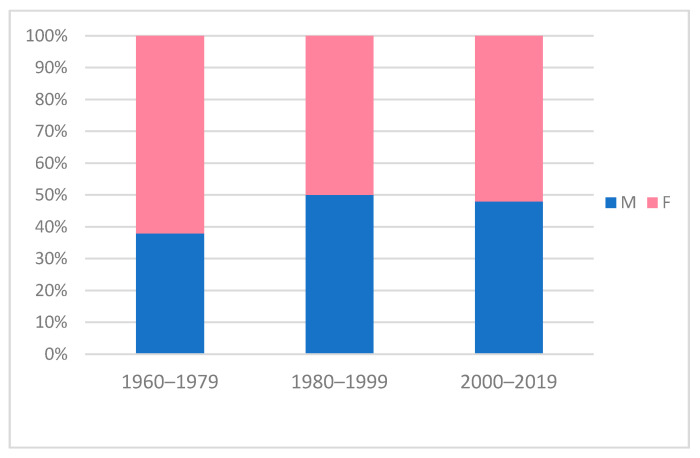
Graph showing gender prevalence of patients with CPC by decades. Males are depicted in blue, females in pink. In the past some authors observed that this condition was more frequent in females [4,10]. However, the M:F ratio showed a trend to 1:1 during the last decades.

**Table 1 children-09-00147-t001:** Characteristics of the studies included in the analysis, sorted in order of publication year. CEBM: Center for Evidence Based Medicine level of evidence; mCMS: modified Coleman Methodology Score. *: in “Patients” section we reported the number of included patients according to the inclusion criteria and the total of patients described by the author/s. For example, Gibson and Carroll described 27 patients in their study, but only 15 of them met the inclusion criteria.

Author and Year	Study Design	Patients *	Details	CEBM	mCMS
Alldred [14] 1963	Case series	9/9	4 conservative 1 excision 1 local autograft 3 autograft from donor site + pin	4	16
Gibson and Carroll [1] 1970	Case series	15/27 4 bilateral	6 conservative (1 bilateral) 2 graft (2 bilateral) 6 graft + pin (1 bilateral) 1 graft + plate	4	7
Owen [15] 1970	Case series	33/33	13 conservative 4 excision 7 autograft from donor site 8 autograft from donor site + pin 1 graft with beef bone peg	4	16
Wall [8] 1970	Case series	5/5	5 conservative	4	4
Ahmadi and Steel [16] 1977	Case series	5/5	3 conservative 1 cancellous autograft + pin 1 graft + pin	4	12
Manashil and Laufer [17] 1979	Case series	3/3	3 conservative	4	0
Toledo and MacEwen [30] 1979	Case series	10/10	6 conservative 4 autograft from donor site + pin	4	25
Quinlan et al. [31] 1980	Case series	4/4	1 conservative 2 autograft from donor site 1 autograft from donor site + wire	4	17
Schnall et al. [32] 1988	Case series	6/6	1 autograft from donor site + pin 5 autograft from donor site + plate	4	24
Grogan et al. [33] 1991	Case series	8/8	8 local autograft + bone suture	4	40
Lorente Molto et al. [34] 2001	Case series	6/6 1 bilateral	1 conservative 5 autograft from donor site + pin (1 bilateral)	4	22
Gomez-Brouchet et al. [35] 2004	Case series	5/5	5 autograft from donor site + pin	4	20
Ettl et al. [36] 2005	Case series	3/3	3 autograft from donor site + pin	4	32
Persiani et al. [7] 2008	Retrospective comparative study	17/17	1 pin 4 plate 7 autograft from donor site + pin 5 autograft from donor site + plate	3b	26
Currarino et al. [37] 2009	Case series	4/4	1 conservative (indicated surgery, lost at F.U.) 1 plate 2 local autograft + plate	4	21
Chandran et al. [29] 2011	Retrospective comparative study	12/12 1 bilateral	2 conservative 5 autograft from donor site + pin 5 autograft from donor site + plate (1 bilateral)	3b	17
Di Gennaro et al. [38] 2016	Case series	26/27	7 conservative 15 autograft from donor site + pin 3 allograft + pin 1 allograft + plate	4	26
Studer et al. [39] 2017	Case series	7/7 1 bilateral	7 autograft from donor site + plate (1 bilateral)	4	50
Giwnewer et al. [40] 2018	Case series	3/3	3 autograft from donor site + plate	4	37
Haddad et al. [41] 2019	Case series	3/4	3 Masquelet technique	4	34
Kim et al. [13] 2020	Retrospective comparative study	47/47 2 bilateral	23 conservative 3 bone suture 5 local autograft + bone suture 1 allograft + bone suture 7 autograft from donor site + plate 4 local autograft + plate 3 allograft and local autograft + plate 1 N/S graft + plate	3b	41

**Table 2 children-09-00147-t002:** Details by groups of surgical procedure. The 164 surgical procedures were divided by type of treatment without distinction for the type of graft eventually used. Data about age at treatment, immobilization, follow-up, nonunion and complication rate are showed in details for each group. BS: bone suture; N/S: non specified; n.: number. *: as described in the text, nonunion was considered a major complication in all groups except in the excision one, since the latter did not pursue healing of the pseudarthrosis site. **: excision group is excluded calculating this prevalence.

	Patients (n.)	Procedures (n.)	Mean Age at Treatment (Years, Range)	Mean Weeks of Immobilization (Range)	Mean Weeks of Consolidation (Range)	Mean Months of Follow-up (Range)	Nonunions (by Procedures %)	Complications (Minor + Major, by Procedures %)	Post-Op Symptoms (by Procedures %)
Excision	5	5	16.0 (1 patient)	N/S	N/S	36.0 (36–36)	5 (100.0%)	0 (0 + 0) * 0.0% *	5/5 (100.0%)
Fixation	9	9	6.0 (4.5–8.0)	6.5 (6.5–6.5)	8.0 (8–8)	58.0 (24–120)	2 (22.2%)	2 (0 + 2) 22.2%	0/6 (0.0%)
Graft alone	13	15	5.6 (2.0–6.5)	N/S	N/S	48.0 (12–84)	1 (6.7%)	2 (1 + 1) 13.3%	1/16 (6.3%)
Graft + BS	14	14	2.6 (0.8–6.0)	6.0 (6.0–6.0)	N/S	94.5 (24–168)	3 (21.4%)	4 (1 + 3) 28.6%	0/8 (0.0%)
Graft + pin	65	67	6.3 (1.5–16.6)	10.1 (6.0–18.6)	10.2 (4–24)	34.9 (2–214)	9 (13.4%)	20 (9 + 11) 29.9%	1/60 (1.7%)
Graft + plate	46	49	7.0 (1.5–16.0)	4.3 (2.0–11.4)	23.7 (8–100)	50.2 (3–120)	5 (10.2%)	8 (2 + 6) 16.3%	3/28 (10.7%)
Masquelet	3	3	8.8 (7.4–9.5)	6.0 (6.0–6.0)	42.0 (42–42)	42.4 (24–120)	0 (0.0%)	2 (0 + 2) 66.7%	0/3 (0.0%)
N/S	1	2	N/S	N/S	N/S	N/S	N/S	N/S	N/S
Total	156	164	5.9 (0.5–16.6)	7.1 (2.0–18.6)	21.2 (4–100)	46.1 (0–214)	25 (15.4%)	38 (13 + 25) 23.9% ** (8.2% + 15.7% **)	10/126 (7.9%)

**Table 3 children-09-00147-t003:** Details by type of graft. The 145 surgical procedures in which graft was used were divided into the following groups without distinction for type of hardware eventually implanted. “Local autograft”: only moreselized fragments of bone from the pseudarthrosis site; “autograft from donor site”: cortico-cancellous bone taken from iliac crest, tibia and/or ribs; “allograft”: any product from bone bank; “others”: provenience of grafts was not clearly specified.

	Procedures (n.)	Nonunions (by Procedures %)	Complications (Minor + Major)
Local autograft	20	3 (15.0%)	4 (1 + 3)
Autograft from donor site	102	11 (10.8%)	24 (10 + 14)
Allograft	9	4 (44.4%)	5 (1 + 4)
Others	14	0 (0.0%)	1 (1 + 0)
Total	145	18 (12.4%)	34 (13 + 21)

## Data Availability

Data available on request due to restrictions. The data presented in this study could be available on request from the corresponding author. The data are not publicly available due to national privacy regulations.

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
