# Peer review of "Congenital Pseudarthrosis of the Clavicle in Children: A Systematic Review"

_children, 2022, doi:10.3390/children9020147_

Round 1

Reviewer 1 Report

Dear authors,

Thank you for the opportunity to review this interesting manuscript analyzing the treatment difficulty of congenital pseudarthrosis of the clavicle. Especially due to the rarity of this condition, various types of publications can be found on this topic. This makes a structured analysis of these studies and a possible context for the treatment recommendation even more important. The content of the manuscript is relevant to the readership and falls within the scope of the journal.

Even though I enjoyed reading the paper, several general questions emerged upon reading.

There are some aspects of the submitted manuscript that need to be revised, and my suggestion is to accept the paper after minor revision. Please consider the following comments and suggestions when amending your manuscript.

Overall recommendation: Accept after minor revision.

General comments:

  • The scientific quality of the work is good. The study is of high relevance considering the small number of publications on this topic. The study approach and design, including the applied methods, are inappropriate on some points and are not presented in sufficient detail.
  • The presentation of the study is suitable. The manuscript is overall well-written in a clear language, and the body of the manuscript is generally well-structured. However, moderate editing of English language and style is required, and some passages should be rearranged for clarity of presentation and to improve readability (see Specific comments).
  • The title is specific and reflects the content of the manuscript.
  • The abstract is brief and indicates the purpose and significance of the study. However, a small adaption is recommended to support the conclusions drawn (see Specific comments).
  • The introduction section provides a comprehensive overview of the subject.
  • The methods section clearly describes the selection and evaluation process but has some shortcomings concerning the statistical methods applied (see Specific comments).
  • In the results section, the findings are clearly presented in general. However, the presentation of the tables appears not well-formatted (see Specific comments).
  • The discussion section is mostly stringent and addresses the relevant aspects of the findings of the study. However, the discussion of the study’s limitations should be extended (see Specific comments).
  • The conclusion section is brief and concise, but the conclusions drawn should be relativized (see Specific comments).
  • The 51 references seem to be relevant. However, there are formatting irregularities (E.g., page numbers, volumes) that need to be corrected, and the authors are encouraged to provide the DOI numbers of each reference.

  • The semicolon appears more than 35 times in the running text, sometimes in inappropriate positions. To improve readability, it should be used very restrictively.
  • The authors state in the introduction section that the decision for conservative or surgical treatment should be facilitated by the analysis performed in the review. Unfortunately, their criteria for indication of surgical treatment remain unclear in the results section. On the other hand, a recommendation for surgery is made based on these biased results.
  • It is pointed out that the study has many limitations. However, relevant limitations are described in a single sentence. A lack of raw data, small number of patients in each study, exclusion of non-English articles are only a few further issues that should be discussed in addition.
  • The headings of the tables and figures are often not very informative. They need to provide enough information about the content without reading the text.
  • Please consistently use a point instead of a comma when providing decimals.

Specific comments:

Abstract

  • l. 13: Please delete the word all as there may be even more relevant studies not considered in this review.
  • ll. 12-14: Please rephrase the sentence relativizing the aims of the study.
  • ll. 21-22: Please specify the rate and provide the corresponding p-value(s).
  • ll. 22-24: The authors should specify and relativize the conclusions drawn without valuing their work. Please delete redundant information as “very rare deformity.”  Please rephrase.
  • l. 24: Information provided in this review was analyzed by the authors which should be clearly distinguished from their opinion.
  • ll. 25-26: Please specify the rate of complications.

Keywords

  • Please use a uniform font 

Introduction

  • ll. 40-43: Please specify the amount/number instead of using “several” and “couple.”
  • l. 43: Redundant sentence, please delete.
  • l. 50: “Painless” or “without pain” may be a better expression instead of “not painful.”

Materials and Methods

  • l. 67: Please rephrase the opening part of the sentence.
  • ll. 103, 106, 107: Grades mentioned here are not analyzed in this review; please clarify.
  • l. 108: Please avoid using informal short forms.
  • l. 116: It remains unclear if raw data were analyzed using the mentioned scores or if these scores have been compared between the studies.
  • l. 117: No information was provided regarding the definition of the confidence interval, the normal distribution or testing for normal distribution, and when which tests were used.

Results

  • ll. 131-132: Please rephrase this part; it remains unclear.
  • Table 1: Please specify the sorting of the data.
  • ll. 147-148: Please use the past tense in this sentence.
  • l. 159: “of these” may be a better expression.
  • ll. 160-162: When listing diseases, a uniform formulation and numbering is appropriate.
  • l. 166: Please clarify why this patient was excluded.
  • l. 168: Please use uniformly “cases” or “patients.”
  • ll. 174-176: Please state the p-value.
  • l. 179: Please indicate the percentage as before.
  • Table 2: No relevant information can be drawn from the title; please modify. Is the mean age at treatment reported in years?
  • ll. 192, 198: Please specify the coefficient used.
  • ll. 195, 196: In these sentences, a more appropriate wording would be “patients treated with autograft” or “suffering non-union.”
  • ll. 203-205: Please rephrase or restructure this sentence.
  • Table 3: Please rephrase the title, providing more information. Please add percentages while presenting complications.
  • l. 212: Please provide percentages uniformly when reporting proportional results.
  • ll. 215-216: “children’s cohort” and “surgical treatment with” may be more appropriate.
  • ll. 217-218: This sentence does not provide adequate information; please rephrase, specify, or remove.
  • ll. 218-219: Which procedures were evaluated at which mean follow-up?

Discussion

  • l. 231: Please remain neutral or describe that it was not found. The use of self-directing pronouns as “we” implies an own opinion.
  • l. 238: Please rephrase “clinical point of view” to, e.g., “clinically.”
  • l. 246: Please specify that these patients are adults.
  • l. 256: Please specify the percentage of “very low.”
  • ll. 266-273: This section describes the meticulous work that was certainly put into this study. However, the description relativizes the relevant limitations and does not contribute relevantly to the discussion. Please delete.
  • l. 279: Please rephrase to “level of evidence. “

Conclusions

  • ll. 290-291: “The surgical treatment achieved…” may be an improved wording.
  • l. 291: “Due to the rarity of the disease” may be a proper beginning of the sentence followed by “this systematic review.”

Author Response

Dear Reviewer, 

We gladly thank you for your time and consideration. We carefully considered every suggestion and extensively revised the text trying to follow your recommendations as best as we could. 

Title: We revised the title as requested to the Editor.

Abstract: We modified the text, making it less redundant as requested by revisor 1 and specifying missing rates and values. We corrected the overall union rate (84.6% to 87.4%), because we reported the percentage including excision, which, as explained later in the text, is a very old treatment option that does not pursue healing of the fracture. In our opinion the abstract is now clearer.

Introduction: Minor changes were accepted; we tried to better focus the rationale of the study as requested by a revisor.

Materials and Methods: Minor changes were accepted. This section was partially rewritten as suggested by editor. The role of Clavien-Dindo-Sink (CDS) in this analysis was better exposed in the following sections of the text. We just would highlight that we carefully analyzed if the authors used clinical derived and/or patient-derived scores to test their patients and which score was used. Here we report some examples which were found in few studies. Anyway, we could not apply them on raw data.

Results: We wrote more detailed highlights and accepted all minor changes suggested. One revisor was understandably surprised we did not asses the refracture rate. We better explained in the text (and would point out here) that none of the authors of the 21 studies reported any case of fracture of the treated clavicle in the postoperative period. As already visible in Supplementary and in the old version of the text, the screening of case report of CPC (19 surgical cases) allowed us to retrieve just one case of pure refracture in the pseudoarthrosis site 2 months after surgery, which required surgical treatment. We hope we clarified this point. We will be very glad if someone report us cases of refracture we did not consider in this analysis.

Discussion: We accepted all minor revisions. As suggested, we frankly pointed out all the main issues of the study deleting most apologetic sounding sentences.

Conclusion: All minor revision accepted.

Again, thank you for your time and consideration.

Respectfully,

Dr Alessandro Depaoli

Dr Paola Zarantonello

Dr Giovanni Trisolino

Reviewer 2 Report

Presented here manuscript entitled „Congenital Pseudarthrosis of the Clavicle in Children.  A Systematic Review with Pooled Analysis and 3 Meta-Regression. ” is the review of CP of the clavicle which, according to the authors claims, has not been published before. The structure of the article is rather consistent and coherent, and the language used is adequate. The purpose of the study is clearly defined, however the rationale for it are blurred in my opinion. The number of publications assessed in this study was rather high. Unfortunately, most of the studies which met the inclusion criteria presented rather poor methodological quality. Moreover, one of the most important factors in the assessment of the treamtnet of pseudoarthroses (refracture rate!) was skipped in this article. 
In summary, this article has a potential to be published in MDPI Children, however, in my opinion,  it needs major corrections before publishing. Therefore, I would like the authors of this article to address below comments:
Intorudcion is lacking the rationale for this study. Please, rewrite the introduction to advocate the aim of this study. 
To make it easy to read, the abbreviations used in tables should be spelled out in the tables descriptions.
line 43: However a genetic etiology seems unlikely[11]. Please remove, or rephrase.
lines 60-61: Please, add (e.g. to the supplementary materials) all the terms which were used for the sake of finding the articles. 
All data presented in this study lack the information on the rate of refracture!Please, add these numbers and add the details concerning refractures in methods, respectively. Please add a relevant comment in the discussion. This point is absolutely essentialn
I would like to commend authors for adding table 3, which clearly shows the non-union rates and possible complications by main procedures used for the treatment of clavicle congenital pseudoarthrosis.
lines 255-257: again, the rate of refracture should be analysed and summarised.

Author Response

Dear Revisor,

Thank you for your time and consideration. We carefully considered every suggestion and extensively revised the text trying to follow your recommendations as best as we could. We hope you can see that we not only used the “Track Changes” function on MS Word. However, we present here a point-by-point reply.

  • Introduction is lacking the rationale for this study. Please, rewrite the introduction to advocate the aim of this study.

As suggested, we better focus the rationale of the study.

  • To make it easy to read, the abbreviations used in tables should be 
    spelled out in the tables descriptions.

We spelled all abbreviations in tables descriptions.

  • line 43: However a genetic etiology seems unlikely[11]. Please remove, 
    or rephrase.

We removed in line 43 as suggested.

  • lines 60-61: Please, add (e.g. to the supplementary materials) all the terms which were used for the sake of finding the articles.

We added all the terms we used in text. As synonyms, we intended e.g. “pseudarthrosis” and “pseudo-arthrosis” or “nonunion” and “non-union”.

  • All data presented in this study lack the information on the rate of 
    refracture! Please, add these numbers and add the details concerning 
    refractures in methods, respectively. Please add a relevant comment in the discussion. This point is absolutely essential.

We are grateful you made clear to us that we did not assess the refracture rate. We better explained in the text that none of the authors of the 21 studies that met criteria reported any case of fracture of the treated clavicle in the postoperative period. As already visible in Supplementary, the screening of case report of CPC (19 surgical cases) allowed us to retrieve just one case of pure refracture in the pseudoarthrosis site 2 months after surgery, which required surgical treatment. We hope we clarified this point.

  • I would like to commend authors for adding table 3, which clearly shows the non-union rates and possible complications by main procedures used for the treatment of clavicle congenital pseudoarthrosis.

We specified rates (%) for complications in Table 3 as suggested.

  • lines 255-257: again, the rate of refracture should be analysed and 

We have already clarified the request.

Again, thank you for your time and consideration.

Regards,

Dr Alessandro Depaoli

Dr Paola Zarantonello

Dr Giovanni Trisolino